# Trabectedin and Lurbinectedin Modulate the Interplay between Cells in the Tumour Microenvironment—Progresses in Their Use in Combined Cancer Therapy

**DOI:** 10.3390/molecules29020331

**Published:** 2024-01-09

**Authors:** Adrián Povo-Retana, Rodrigo Landauro-Vera, Carlota Alvarez-Lucena, Marta Cascante, Lisardo Boscá

**Affiliations:** 1Instituto de Investigaciones Biomédicas Alberto Sols-Morreale (CSIC-UAM), Arturo Duperier 4, 28029 Madrid, Spain; rlandauro@iib.uam.es (R.L.-V.); clucena@iib.uam.es (C.A.-L.); 2Department of Biochemistry and Molecular Biomedicine-Institute of Biomedicine (IBUB), Faculty of Biology, Universitat de Barcelona, 08028 Barcelona, Spain; martacascante@ub.edu; 3Centro de Investigación Biomédica en Red de Enfermedades Hepáticas y Digestivas (CIBEREHD), Instituto de Salud Carlos III (ISCIII), 28029 Madrid, Spain; 4Centro de Investigación Biomédica en Red de Enfermedades Cardiovasculares (CIBERCV), Institute of Health Carlos III (ISCIII), 28029 Madrid, Spain

**Keywords:** trabectedin, lurbinectedin, macrophages, lymphocytes, ecteinascidins, combined therapies, innate immunity, adaptative immunity, molecular oncology

## Abstract

Trabectedin (TRB) and Lurbinectedin (LUR) are alkaloid compounds originally isolated from *Ecteinascidia turbinata* with proven antitumoral activity. Both molecules are structural analogues that differ on the tetrahydroisoquinoline moiety of the C subunit in TRB, which is replaced by a tetrahydro-β-carboline in LUR. TRB is indicated for patients with relapsed ovarian cancer in combination with pegylated liposomal doxorubicin, as well as for advanced soft tissue sarcoma in adults in monotherapy. LUR was approved by the FDA in 2020 to treat metastatic small cell lung cancer. Herein, we systematically summarise the origin and structure of TRB and LUR, as well as the molecular mechanisms that they trigger to induce cell death in tumoral cells and supporting stroma cells of the tumoral microenvironment, and how these compounds regulate immune cell function and fate. Finally, the novel therapeutic venues that are currently under exploration, in combination with a plethora of different immunotherapeutic strategies or specific molecular-targeted inhibitors, are reviewed, with particular emphasis on the usage of immune checkpoint inhibitors, or other bioactive molecules that have shown synergistic effects in terms of tumour regression and ablation. These approaches intend to tackle the complexity of managing cancer patients in the context of precision medicine and the application of tailor-made strategies aiming at the reduction of undesired side effects.

## 1. Introduction

Oceans and seas constitute 71% of the Earth’s surface and account for 90% of biodiversity on our planet. Marine ecosystems are composed of complex communities of animals, plants, fungi, and microorganisms such as bacteria, protozoa, algae, and chromists [1]. Hence, the marine biosphere is rising as a fundamental potential source of bioactive molecules [2].

Thousands of marine natural products with biological therapeutic relevance are identified every year; in 2017, 1490 novel compounds were reported in 477 articles and 1544 were reported in 2018, which were vastly documented in 469 publications [3].

There is a growing interest within the pharmaceutical field for drug screening, discovery, and development in aquatic ecosystems due to the secondary metabolites that are generated by marine organisms.

## 2. Ascidians as a Source of Bioactive Molecules: *Ecteinascidia turbinata*

Ascidians are ancestral marine urochordates and tunicates that are considered filter-feeders [4]. This is the reason why they are considered pollution indicators and present unique characteristics in the animal kingdom; these organisms produce alternative proteins, such as specific oxidases and phytochelatins, and synthesize cellulose.

This enormous family is composed of more than 3000 different species whose reproduction is both sexual and asexual and, up-to-date, these organisms have directly been identified as a source of more than 1200 bioactive molecules [5].

Thus, ascidians have drawn the attention of the biomedical field due to their ability to synthesize secondary metabolites. Attending to the chemical nature and molecular structure of these biomolecules, there are three main groups: alkaloids, peptides, and polyketides [5,6,7].

Alkaloids are the most prominent family of compounds that exert antimicrobial [6] and anti-tumour activities [1,5,8]. Within this group it is relevant to mention saframycins, jorumycins, renieramycins, and ecteinascidins that share structural similarities within the bis-tetrahydroisoquinoline chemical moieties [9]. They inhibit essential kinases that regulate the cell cycle, such as protein kinase B (PKB) and cyclin-dependent kinases (CDKs), and alter the mitochondrial inner membrane potential [5]. Trabectedin and lurbinectedin, the anti-tumour compounds marketed by PharmaMar, belong to this category. Peptides from two to eighteen amino acids constitute a smaller subset (5% of bioactive molecules) distributed into linear peptides, cyclic peptides, and depsipeptides (peptides composed by ester and amide bonds). Finally, a third group has been identified, polyketides, which are complex molecules built from simple carboxylic acids and synthesized by polyketide synthetases [5,10].

This is a simplified overview as, not only are these molecules synthesized by ascidians, but their associated symbiont microorganisms have a pivotal role in the production of these defensive molecules that protect these marine creatures from their natural predators [5,7,11,12].

Ascidians show enhanced cellular plasticity within the Chordate phylum, and for this reason, are used as regenerative biology models [13].

*Ecteinascidia turbinata* is a tunicate that normally inhabits the Caribbean Sea, Gulf of Mexico, Bermuda, East Coast of Florida, and it has been seen in the Mediterranean Sea in the warmest periods [4] (Figure 1). This organism and a symbiont, *Candidatus Endoecteinascidia frumetensis*, are the natural sources of trabectedin [6,11], and it was the first ascidian compound with an anti-tumour activity to receive both EU (EMA) and US (FDA) approval [4].

## 3. Trabectedin and Lurbinectedin Molecular Structures

Ecteinascidins’ structures were first reported in 1987 by Rinehart et al., although the anti-tumour activity of these compounds was reported in 1969 from tunicate total extracts; however, due to the limited availability of the primary source, it took almost two decades to identify the molecular structure of the compounds that exerted the anti-tumour activity [14].

Trabectedin (ET-743 or TRB) was first isolated in 1990 and its X-ray crystallographic structure was resolved in 1992 [13]. It is a tetrahydroquinoline alkaloid with a molecular weight of 761.81 g/mol and a complex chemical structure composed of three fused tetrahydroisoquinoline rings (subunits A–C): a mono-bridged pentacyclic skeleton of two tetrahydroisoquinoline rings (subunits A and B) linked by a 10-member lactone bridge through a benzylic sulphide linkage attached to an additional ring by a -spyro ring to a third tetrahydroisoquinoline structure (Figure 2A). It is semi-synthetically produced and currently marketed as Yondelis^®^ (Madrid, Spain) [15]. The complete synthesis of this compound by Ma and Chen’s group has recently been reviewed by Gao et al. [9], along with other tetrahydroisoquinoline alkaloid compounds.

Lurbinectedin (PM01183 or LUR) is a structural derivative of the former molecule. TRB and LUR differ in the C-subunit; TRB presents a tetrahydroisoquinoline (circled in blue) which is replaced by a tetrahydro-β-carboline (circled in red; Figure 2B) [16,17]. As a consequence, there are important pharmacodynamic and pharmacokinetic modifications [17]; LUR exhibits a distribution volume four times lower than TRB and exhibits a three-fold higher tolerance dose (MTD), presenting a distinct profile [18]. The LUR molecular structure is slightly larger. The molecular weight for PM01183 is 784.87 g/mol, and it is commercialized as Zepzelca^®^ (Madrid, Spain).

**Figure 2 molecules-29-00331-f002:**
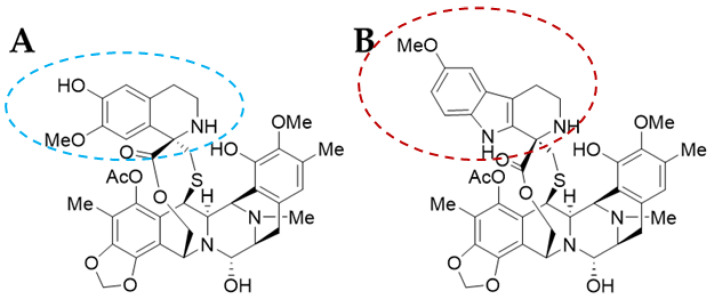
Molecular structure of trabectedin [14] (**A**) and lurbinectedin [18] (**B**).

## 4. Trabectedin and Lurbinectedin Uses in Oncology

TRB is indicated, in combination with pegylated liposomal doxorubicin, for patients with relapsed platinum-sensitive ovarian cancer [19,20,21], as well as for the treatment of advanced soft tissue sarcoma in adults [22,23,24,25] when ifosfamides and anthracyclines have failed [19,20].

TRB is applied for soft tissue sarcoma (STS) [26] and is applied in a phase III study of mesenchymal chondrosarcoma [27] as well as in a phase II study of extraskeletal myxoid chondrosarcoma. It is also used in non-operable liposarcomas and leiomyosarcomas [28,29,30,31,32,33,34] and it is applied for metastatic synovial sarcoma [35,36].

Recently, monocyte to lymphocyte ratio was demonstrated to be a prognostic tool in the treatment of STS with trabectedin. This ratio could be easily applied in clinical practice to assess TRB efficacy [37].

Lurbinectedin (LUR) was approved by the FDA in June 2020 after a phase II/III trial for the treatment of small cell lung carcinoma [38]. Additional assays were conducted for the treatment of ovarian cancer, breast cancer (ongoing in patients of BRCA1 loss of function), sarcoma, and acute myeloid leukaemia [39,40,41,42,43].

## 5. Mechanism of Action of Trabectedin and Lurbinectedin

Several mechanisms of action have been described for TRB and LUR. The most extensively documented mechanisms are related to their role as DNA-binding agents and transcriptional modulators. Still, there is mounting evidence suggesting that, apart from the well-known molecular mechanisms that are affected in tumoral cells (described below), the impact on the immune system compartment is of pivotal importance in the outcome of patients who are treated with these ecteinascidins [44,45].

### 5.1. TRB and LUR Act as DNA Intercalating Agents and Transcriptional Regulators

Both compounds bind to guanines in the DNA minor groove, specifically to the exocyclic N_2_ amino moiety through an in situ iminium intermediate that is formed by the dehydration of the carbinolamine that is located in the A subunit. Thus, TRB and LUR act as intercalating DNA agents [45]. The DNA-TRB or LUR adduct is additionally stabilised by existing van der Waals interactions and hydrogen bonds between the A and B subunits. This covalent, newly formed interaction induces a DNA torsion towards the major groove; this feature seems to be unique to these families of compounds [17,46,47].

Combinatorial chemical substitutions revealed that the carbinolamine moiety is relevant to the pharmacological activity of these molecules since derived compounds (ET-745) lacking this functional group fail to bind to DNA. Both anti-tumour drugs induce transcription-dependent stress and genomic instability [48]. In addition, this interaction with the DNA interferes with the transcription of genes whose promoter regions contain CG-rich sequences. Moreover, this transcriptional regulation is implemented by the dephosphorylation, ubiquitination, and degradation of the RNA polymerase II (Pol II) on the DNA template [49].

### 5.2. TRB and LUR Affect Homologous Recombination (HR) and Nucleotide Excision Repair (NER) DNA Repair Mechanisms

The main DNA repairing mechanism that is affected by TRB and LUR is nucleotide excision repair (NER). When there is a lack of this process, the cytotoxic capacity of these molecules over tumour cells is diminished [50]. If the affected mechanism is homologous recombination (HR), as it occurs in decreased expression of BRCA1/2, the reported cytotoxic effect is higher. TRB and LUR inhibit both NER and HR in tumoral cells [51]. Two of the principal components of the NER mechanism are XPG (Xeroderma pigmentosum group G) and XPF (Xeroderma pigmentosum group F) endonucleases that cleave the damaged DNA double-strand and correct the lesion [52]. In vivo studies in the yeast *S. pombe* showed that Rad13, a homolog protein, forms a tertiary complex, Rad13-DNA-TRB, that induces double-strand breaks, leading to the activation of programmed cell death processes [53].

### 5.3. TRB and LUR Affect Transcription, Cell Cycle, and Induce Apoptosis in Tumour Cell Lines

The steric hindrance induced by the tertiary complex prevents transcription factor binding to conserved consensus DNA sequences where there is an enrichment in GC. A dose-dependent binding inhibition has been reported at low micromolar doses (50–300 µM) for TBP, E2F, SRF, and CCAAT transcription factors by gel shift assays, and at even lower concentrations for NF-Y (10–30 µM). TRB induced a decrease of nucleosomes at 100 nM [54,55].

NF-Y is a central transcription factor that mediates the activation of the human gene that codes for P-glycoprotein or MDR-1, which recruits histone acetyltransferase PCAF to the MDR-1 promoter. TRB abrogates its transcriptional activation [56,57] and in doing so, it prevents ABCB1 channel activity, and therefore, it avoids the multidrug resistance that is associated with the overexpression of this protein in tumour cells [58,59,60].

In the low nanomolar range, TRB inhibits B2 cyclin transcription which might explain G_2_ cell cycle blockade [55,61]. It activates non-dependent P53 apoptosis and produces a cell cycle blockade in the late S and G_2_-M phases [53,61].

These anti-tumour compounds trigger RNA polymerase dephosphorylation and facilitate RNA polymerase II degradation via ubiquitination [16,62,63,64], drastically modulating messenger RNA transcription [16,65].

TRB and LUR induce apoptosis by both the intrinsic and extrinsic pathways in lung cancer A549 cell lines [66], and in MCF-7 and MDA-MB-453 breast cancer cells [67] in a time- and concentration-dependent fashion. It has been proposed that lurbinectedin in monotherapy is more effective for relapsed SCLC than other approved therapies [68,69,70].

### 5.4. TRB and LUR Regulate Tumour Microenvironment

It has extensively been reported that these anti-tumour drugs impact the tumour microenvironment (TME), target human tumour-associated macrophages (hTAM) [46,71,72,73], and inhibit the transcription of pro-inflammatory cytokines such as CCL2 (chemokine ligand 2), IL-6 [74], VEGF [65], CCL3, CCL7, and CCL14 [53,75]. TRB and LUR are known to modulate the immune response within the tumour microenvironment by specifically targeting mononuclear phagocytes [44,65,73,76,77]. Furthermore, it has recently been shown that TRB and LUR modulate the macrophage electrophysiology and polarisation state towards a proinflammatory-like (M1) activation state in quiescent macrophages, suggesting that TAMs pro-inflammatory re-education occurs in murine peritoneal rodent macrophages [78]. Moreover, LUR effectively eliminates both cancer cells and cancer stem cells in preclinical models of uterine cervical cancer [79]. Human TAMs are functionally inhibited and depleted by TRB, which improves the anti-tumour adaptative response to anti-PD-1 therapy [80].

### 5.5. TRB and LUR Affect the Human Immune System

Both TRB and LUR exert a direct impact on all the immune cell subsets, which probably contributes to the therapeutic actions of these drugs. Nevertheless, adverse effects have occasionally been observed in oncological patients, constituting an exclusion criterion for patients undergoing treatment with these anti-tumour drugs [75,81,82,83]. At this point, the development of prognostic biomarkers associated with the appearance of adverse effects after treatment with these drugs is a relevant area of research. Additionally, both drugs have been proposed to be applied in combination with immune checkpoint inhibitors, along with a plethora of specific targeted therapies, as addressed in Section 7.

#### 5.5.1. Impact on Phagocytes/Myeloid Compartment (Neutrophils, Monocyte/Macrophages, DCs)

The most common adverse effect of TRB or LUR administration is neutropenia, which is reported in one-third of cancer patients undergoing these treatments, and if it is severe, it constitutes a motive for withdrawal. Both drugs target the mononuclear phagocytic system, specifically, monocytes and macrophages. They can inhibit cytoskeleton dynamics and motility, phagocytosis, and efferocytosis, and trigger apoptosis, as well as the recruitment of monocytes to the tumour site and induce apoptosis [65,84,85,86].

There are no currently available studies on dendritic cells regarding TRB and LUR. It would be very interesting to evaluate the dual role of these cells in tumour immunity [87,88,89,90,91,92]. However, TRB antitumoral activity was assessed in an orthotopic xenograft murine model bearing a doxorubicin-resistant follicular dendritic cell sarcoma derived from a patient and it was concluded that this tumour was slightly sensitive to TRB, but it was not statistically significant [93].

It has been proposed, extensively studied, and reviewed that these anti-tumour molecules exert “tropism” for hTAM [44]. They inhibit angiogenesis by inhibiting the expression of VEGF, PDGF, FGF, and metastasis by regulating MMPs, and abolish the immunosuppression that is established within the TME. In this sense, LUR has been identified as an inhibitor of myeloid suppressor cells, both in vivo and in vitro [79,94].

The impact of TRB and LUR on human macrophages has been extensively reported: these antitumoral compounds induce programmed cell death in sensitive macrophages [86] and, in those that retain viability, it favours a pro-inflammatory-like activated state. These compounds upregulate HLA (MHC class I and II) transcripts, glycolysis, NF-κB, and P38 proinflammatory pathways, and induce mitochondrial biogenesis. Additionally, both antitumoral drugs activate PPP and increase NADPH-oxidase-dependent ROS production as well as O_2_^−^ generation and induce a rupture of the TCA cycle at MDH and IDH, favouring the metabolic HIF-1α stabilisation. This metabolic reconfiguration leads to the canonical hMφ proinflammatory activation [95]. TNFα and IL-8 are augmented in the supernatant of primary hMφ [96] (Figure 3).

Hence, not only do these ecteinascidins impact the tumour microenvironment and the tumour itself, but they also trigger a proinflammatory activation, at least in in vitro primary human macrophages’ cell culture. Macrophage polarisation is driven by macrophage metabolism, and it regulates the biological function of these innate cells [95,97,98]. The understanding of these processes is crucial to comprehend the interplay between immune cells and the tumour and its microenvironment to design specific targeted therapies to improve oncologic patient outcomes and overall survival, as well as progression-free survival.

#### 5.5.2. Impact on Lymphoid Subsets (T Cells, B Cells, NK Cells, and NKT Cells)

These molecules activate NK cells; TRB triggers direct and NK-mediated cytotoxicity in multiple myeloma [99], and both TRB and LUR exert a cytotoxic effect targeting B cells in Chronic Lymphocytic Leukaemia (CLL) [43,100]. TRB exhibited cytotoxic effects in diffuse large B cell leukaemia [101]. TRB and LUR have been shown to activate CD4^+^ and CD8^+^ T-cells as well, promoting the adaptive anti-tumour immune response, inducing their infiltration in vivo and the proliferation of activated effector T-cells in vitro [43,100,102,103].

Globally, TRB and LUR functionally/mechanistically exert three major roles: these drugs induce apoptosis in tumour cells and stromal supporting cells (TAMs), modulate the TME, and instruct both innate and adaptive immune cells towards an anti-tumour-activated phenotype. Thus, they directly kill tumour cells and prevent the immunosuppressive milieu that is established. These molecules potentiate the anti-tumour immune response to neutralise the tumour.

It has been suggested that the expression of TRAIL-R in the different leukocyte subsets is related to the mechanism of TRB-induced apoptosis and could be useful to explain the differential viability effects on cell viability of each of the myeloid and lymphoid subsets [44,100].

Figure 4 recapitulates the most extensively reported effects of TRB and LUR on tumour cells, the tumour microenvironment, and tumour-supporting cells, as well as immune cell activation, and is supported by experimental evidence [86,96].

Both drugs exert a direct cytotoxic effect in tumour cells by interfering with the transcription machinery and cell cycle and inducing immunogenic tumour cell death. As a result, they inhibit the immunosuppressive milieu that is normally established by the tumour and tumour supporting cells. TRB and LUR downregulate the expression of VEGF and several metalloproteases, preventing both tumour progression and metastasis and simultaneously activating NK-mediated cytotoxicity, T-cell infiltration (in vivo), and macrophage proinflammatory activation (in vitro). They reduce monocyte migration (Figure 4).

Taken together, these data strongly suggest that TRB and LUR elicit a higher immune response through two different paths: these drugs prevent the functional pro-tumoral immune suppression in the TME and favour immune cell activation, which explains tumour regression and overall patient improvement and makes these molecules great candidates for their combination with immunotherapy. Therefore, it is highly relevant to establish and explain how these ecteinascidins modulate the human adaptative immune response since mounting evidence demonstrates that, not only do these drugs induce tumour cell death, but they also instruct the immune system to activate and respond to neutralise the tumour.

## 6. Novel Functional and Molecular Targets for Trabectedin

Trabectedin induces ferroptosis via HIF-1α/IRP1/TFR1 and Keap1/Nrf2/GPX4 in non-small cell lung cancer cells (nSCLC) [104]. This biological process is proven to be essential in macrophage function and is arising as a novel target in oncology that is increasingly becoming a hot topic within the molecular oncology and immunotherapy field. NRF2 and redox biology seem to be regulated by the molecular mechanism of TRB [96,104], although this observation deserves further investigation.

It has additionally been nominated as a potential candidate for drug repositioning in the FDA for type II diabetes treatment by docking. It has been postulated as an α-glucosidase inhibitor with an in vitro IC_50_ of 1.2 ± 0.7 µM, alongside demeclocycline. Nonetheless, this repositioning needs to be further assessed due to its systemic toxicity, hence, a well-justified safety study ought to be conducted [105].

Recently, it has been shown that TRB inhibited therapy induced senescence in tumours by altering glutamine metabolism [106].

## 7. Combination Therapies Involving Trabectedin and Lurbinectedin

There is a pressing need to design combination treatments that may include conventional chemotherapeutic agents, immunologically targeted therapies such as immune checkpoint inhibitors (ICIs), or specific inhibitor molecules that target signaling or metabolic regulators, due to the complexity of tumoral biology and adaptation capacity, as well as resistance generation. In this sense, the field is experiencing a remarkable expansion: globally, TRB is assessed in combination with ICIs (antiPD-1/PD-L1 [107,108,109,110] and/or CTLA-4 [109]), monoclonal antibodies (-mAbs) that may act as either molecular inhibitors or activators, specific inhibitors of molecular targets (PARP [51,111,112,113], MDM2 [114], VEGF [115,116,117], CCR5 [118], m-TOR [119], IGF1-R [120], BCL2 [121], ATM/ATR [122],PPAR-γ [123], and CK-2/CLK2 [124]), recombinant proteins (shTRAIL [125]), topoisomerase inhibitors (irinotecan [126,127,128,129,130,131], topotecan [127], and camptothecin [132]), and immuno-modulatory biomolecules such as L19-mTNF [133] or dexamethasone [134], combined with propranolol [135], a β-adrenergic receptor inhibitor, or Wnt/β-catenin inhibitors [136] (PRI-724). It is combined with physical agents (hyperthermia [137] and radiation [138,139,140,141], among other strategies) as it is shown in Table 1, where it is indicated in which pathologies and cellular or murine models are applied.

LUR has been evaluated in combination with ICIs (anti-PD-L1 and anti-CTLA-4 [152]) and in combination with irinotecan [153,154], ATR [122,155] alone or combined with ATM [156] and PARP [157] inhibitors, anti-VEGF [158] combined with cisplatin [83,159,160], paclitaxel [158], gemcitabine [161], capecitabine [162], doxorubicin [41,163,164], and immunomodulatory biomolecules such as antibody-drug complexes commonly referred to as ADCs (4C9-DM1 that targets c-Kit [165]).

TRB in combination with Anti-AXLxCD3ε has proven to be more effective than TRB alone in sarcoma cells [150]. TNT treatment (talimogene laherparepvec, nivolumab, and trabectedin) has shown to be synergistic against advanced sarcoma [108]. There is an ongoing phase I/II SAINT study using ipilimumab (CTLA-4 inhibitor), nivolumab (PD-1 inhibitor), and trabectedin, as a first-line treatment for advanced soft tissue sarcoma (ASTS) (NCT03138161) [109]. TRB + irinotecan has proven to be effective on a desmoplastic small round cell tumour patient-derived xenograft [130,131], cisplatin-resistant osteosarcoma [129], and rhabdomyosarcoma [128]. TRB + β-blocker propranolol combination has proven to be effective in vitro and ex vivo evaluations in cervical cancer in patient-derived organoids [135].

In the ovarian cancer cell model, TRB + anti-PD-1-mab showed synergistic efficacy [102], favouring the activation of effector CD4^+^ and CD8^+^ T-cells in vivo by the upregulation of IFN-γ and inducing a decrease of immunosuppressive MDSCs and regulatory T-cells [103,142]. Three dose levels of TRB + durvalumab (PD-L1 inhibitor) showed promising efficacy in a phase Ib multicentre trial (TRAMUNE) in relapsed platinum-refractory ovarian cancer [107,166].

A similar approach was conducted in murine osteosarcoma models where TRB inhibited osteosarcoma primary tumour growth and metastasis and enhanced the number of T-cell tumour-infiltrated cells (both CD4^+^ and CD8^+^). TRB induced the overexpression of PD-1 in vivo but it did not in vitro, and Chiara Ratti et al. proved that TRB + anti-PD-1 blocking antibody increased CD8^+^ infiltrating cells and TRB efficacy, whereas anti-PD-1 alone did not reduce osteosarcoma growth. The combination further increased CD4^+^ and CD8^+^ recruitment, shifted CD4^+^ naïve T cells to CD4^+^ effector memory cells, and rendered a higher efficacy compared to TRB alone, preventing osteosarcoma progression. This combination enhanced the expression of CTLA-4, suggesting that it might be a third suitable partner for combined immunotherapy [167].

TRB + everolimus was synergistic in cisplatin and paclitaxel-resistant ovarian clear cell carcinoma cell lines and mice xenografts [119]. TRB + maraviroc (CCR5 antagonist) was effective in classical Hodgkin lymphoma mesenchymal stromal cells [118]. In a phase II clinical trial, TRB + dexamethasone improved safety in pre-treated soft tissue sarcoma patients [134]. TRB + mitotane reduced invasiveness and metastatic processes in adrenocortical carcinoma [145]. TRB + metformin + CB2 emerged as a novel venue for diabetes-associated breast cancer in cell lines and xenograft murine models [144]. TRB + PRI-724 [136] and TRB + RG7112 [114] were effective in human in vitro soft tissue sarcoma cell lines (in MDM2-amplified liposarcoma and fibrosarcoma cell lines [114] and STS cell lines and primary cultures [136]). TRB + radiotherapy combination has been assayed in A549 and HT-29 cell lines [138]. This approach has been identified as beneficial for STS patients, especially when tumour sinkage for symptomatic relief is required [139], and for phase I and phase II clinical trials for patients with myxoid liposarcoma. In the first one, there is an improvement in both safety and antitumoral activity [141]; in the second one, the primary endpoint was not achieved but the combination was well-tolerated and effective in terms of pathological response.

LUR has been combined with several agents as it is shown in Table 2 where, again, it is indicated in which pathologies/cellular or murine models the combinations are applied. LUR has been combined with irinotecan in ovarian cell clear carcinoma cell lines showing synergistic effects [154] and in a case report showing BRCA-mutated platinum-resistant ovarian cancer patients had exceptional clinical responses [153]. LUR + olaparib (PARP inhibitor), in a phase I clinical trial for advanced solid tumours, is feasible and exhibited a disease control rate of 72.6% [157]. LUR + doxorubicin improved safety in a phase III clinical trial of SCLC [164], and in a phase II clinical trial, a benefit was observed in several types of metastatic and unresectable sarcomas [41]. In an expanded phase I clinical trial for advanced endometrial cancer, this combination favoured a better response rate as its duration progressed [163].

LUR + capecitabine was applied in a phase I trial for relapsed metastatic breast cancer (HR+) with promising results [162]. LUR + paclitaxel showed synergistic antitumoral activity and improved safety in a phase I trial in SCLC, breast and endometrial cancer patients, and in combination with bevacizumab (anti-VEGF) for epithelial ovarian cancer [158]. TRB or LUR + VE-821 + KU-60019 (anti-ATR and anti-ATM respectively) combinations were evaluated in ovarian and cervical cell lines and showed higher antitumoral activity, suggesting that this venue provided mechanistic evidence that could have potential therapeutic effects that need to be addressed [122]. For LUR in combination with ICIs: anti-PD-L1 and anti-CTLA-4 showed strong anti-neoplastic effects in osteosarcoma and fibrosarcoma cell lines, and breast cancer and fibrosarcoma murine models [152]. LUR + berzosertib (ATR inhibitor) showed synergy in SCLC in vivo, organoid, and in vitro models [155]. LUR + cisplatin showed promising activity in malignant pleural mesothelioma [160] but it was not feasible in advanced solid tumours due to toxicity issues [83]. TRB + gemcitabine was assessed in the phase I trial for several advanced tumour types showing well-tolerated results with higher antitumoral activity [161]. Finally, LUR + 4C9-DM1-ADC achieved a higher tumour growth inhibition rate compared to LUR alone in mice xenografts bearing human SCLC [165].

Interestingly, it has been proposed that these Ecteinascidins may be combined with specific antibodies forming antibody-drug complexes (ADC) [165] and nanoparticles [168]. These approaches might help to overcome unwanted collateral side effects and improve safety parameters, both locally in the vasculature at the injection site, and systemically.

## 8. Conclusions/Concluding Remarks

TRB and LUR induce apoptosis and immunogenic cell death in tumours through diverse molecular mechanisms that are still being identified.

TRB and LUR function as immune-modulatory drugs, both in the TME and over the innate immune cell compartment, as well as the adaptative compartment, irrespectively of the myeloid or lymphoid origin, although the most extensively characterised is the first one.

Both Ecteinascidins are being assessed in combination with a plethora of molecular targeted inhibitors, monoclonal antibodies, immune checkpoint blockades, as well as classical oncolytic treatments that include physical agents (radiotherapy or hyperthermia). and canonical chemotherapeutical drugs (i.e., irinotecan or topotecan) that synergise with TRB/LUR, enhancing their antitumoral activity and/or their safety profile.

## Figures and Tables

**Figure 1 molecules-29-00331-f001:**
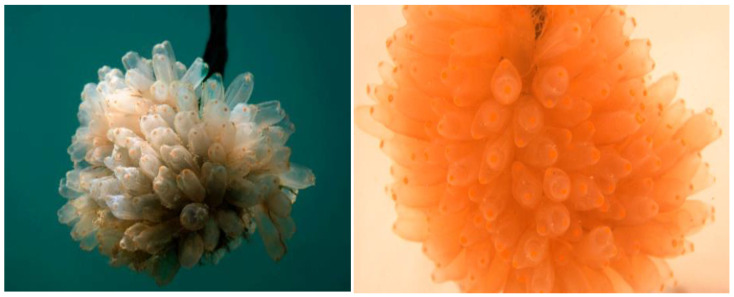
*Ecteinascidia turbinata*, the ascidian source for trabectedin and lurbinectedin (Courtesy of PharmaMar S.A., Madrid, Spain).

**Figure 3 molecules-29-00331-f003:**
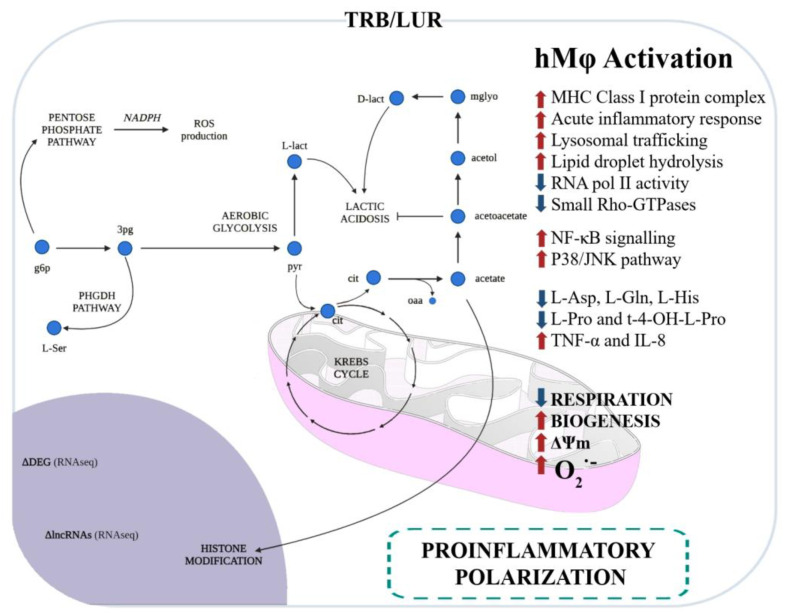
Immunometabolic and functional response of human macrophages to trabectedin and lurbinectedin. Glycolysis and the pentose phosphate pathway (PPP) are favoured, and serine production is predicted by RNAseq and fluxomic approaches [96].

**Figure 4 molecules-29-00331-f004:**
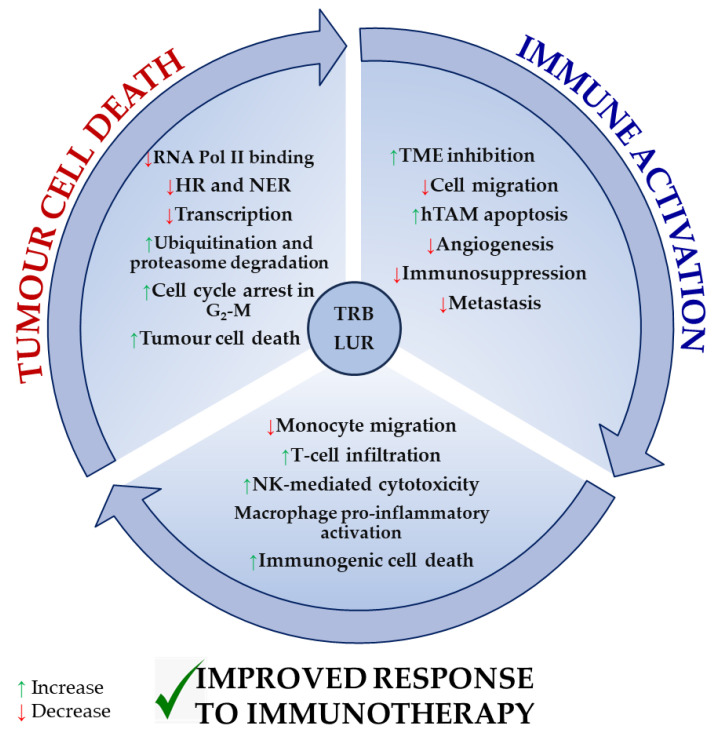
Reported molecular mechanisms for TRB and LUR in tumour cells, the tumour microenvironment, and immune cells. TRB and LUR are DNA intercalating molecules and transcriptional regulators. They impact human TAM biology acting as TME regulators and immunomodulate human immune response and activation.

**Table 1 molecules-29-00331-t001:** Combination therapies involving trabectedin.

Category	Treatment	Co-Treatment Function	Type of Cancer	Ref.
**Monoclonal antibodies** (**mAb**)	TRB + bevacizumab	Anti–VEGF	Partially platinum-sensitive recurrent ovarian cancer	[115]
	TRB + AVE1642	Anti–IGF1R	Ewing sarcoma	[120]
	TRB + VE-821 + KU-60019	Anti–ATR (VE-821) Anti–ATM (KU-60019)	Cervical carcinoma, ovarian carcinoma	[122]
**Immune checkpoint inhibitors** (**ICIs**)	TRB + durvalumab	Anti–PD-L1	Platinum-refractory ovarian carcinoma	[107]
	TRB + avelumab	Anti–PD-L1	Advanced liposarcoma and leiomyosarcoma	[110]
	TRB + nivolumab + talimogene laherparepvec (TVEC)	Anti–PD-1 (nivolumab) Replication within tumours and production of GM-CSF (TVEC)	Advanced previously treated sarcomas	[108]
	TRB + ipilimumab + nivolumab	Anti–CTLA-4 (ipilimumab) Anti–PD-1 (nivolumab)	Advanced soft tissue sarcoma	[109]
	TRB + α-PD-1 mAb		Ovarian cancer	[142]
**Inhibitors**	TRB + olaparib	PARP inhibitor	Breast cancer	[111]
			Advanced and unresectable bone and soft-tissue sarcomas	[112]
			Ewing sarcoma	[113]
			Osteosarcoma, leiomyosarcoma	[143]
	TRB + RG7112	MDM2 antagonist	Soft tissue sarcoma	[114]
	TRB + rucaparib	PARP inhibitor	Soft tissue sarcoma, dedifferentiated liposarcoma	[51]
	TRB + PRI-724	Wnt/β-Catenin inhibitor	Soft tissue sarcoma	[136]
	TRB + ponatinib	Multi-tyrosine kinase inhibitor	Solitary fibrous tumour of the pleura	[116]
	TRB + propranolol	β-adrenergic receptors antagonist	Cervical cancer, ovarian cancer	[135]
	TRB + pioglitazone	PPARγ agonist	Myxoid liposarcoma	[123]
	TRB + topotecan	Topoisomerase I inhibitor	Ovarian clear cell carcinoma	[127]
	TRB + irinotecan	Topoisomerase I inhibitor	Ovarian clear cell carcinoma	[127]
			Rhabdomyosarcoma	[128]
			Cisplatin-resistant osteosarcoma	[129]
			Relapsed desmoplastic small round cell tumour	[131]
			Desmoplastic small round cell tumour	[130]
	TRB + everolimus	mTOR inhibition	Cisplatin-resistant and paclitaxel-resistant ovarian clear cell carcinoma	[119]
	TRB + maraviroc	CCR5 antagonist	Classical Hodgkin lymphoma-mesenchymal stromal cells	[118]
	TRB + metformin + CB-2	Hypoglycemic agent (metformin) MCT4 inhibitor (CB-2)	Diabetes-associated breast cancer	[144]
	TRB + camptothecin	Topoisomerase I inhibitor	Myxoid/round cell liposarcoma, undifferentiated pleomorphic sarcoma	[132]
	TRB + obatoclax	Bcl-2 inhibitor	Malignant pleural mesothelioma	[121]
	TRB + ABT-199	Bcl-2 inhibitor	Malignant pleural mesothelioma	[121]
	TRB + OSI-906	IGF1R inhibitor	Ewing sarcoma	[120]
	TRB + silmitasertib	CK2/CLK double-inhibitor	Uveal melanoma	[124]
	TRB + cabozantinib	c-MET/TAM (TYRO3, Axl, MERTK) receptor inhibitor	Uveal melanoma	[124]
**Biological agents**	TRB + FOLFIRI (leucovorin + 5-fluorouracil + irinotecan)	Treatment of colorectal cancer	Colorectal cancer	[126]
	TRB + mitotane	Treatment of adrenocortical carcinoma	Adrenocortical carcinoma	[145]
	TRB + dexamethasone	Glucocorticoid medication	Advanced/metastatic soft tissue sarcoma	[134]
	TRB + gemcitabine	Treatment of advanced pancreatic cancer can disrupt DNA replication and activate the S phase checkpoint	Pancreatic cancer	[146]
	TRB + paclitaxel	Treatment of advanced solid tumours	Advanced solid tumours	[147]
	TRB + docetaxel	Treatment of ovarian and peritoneal cancer	Recurrent/persistent ovarian and peritoneal cancer	[148]
	TRB + enterolactone	Anti-angiogenic activity	Epithelial ovarian cancer	[117]
	TRB + cisplatin	Treatment of malignant pleural mesothelioma	Malignant pleural mesothelioma	[121]
	TRB + carboplatin	Treatment of advanced solid tumours	Advanced solid tumours	[149]
	TRB + shTRAIL	Targets cancer cells to induce apoptosis	Colon cancer	[125]
	TRB + pAXL × CD3ε	Redirects T-lymphocyte cytotoxicity to AXL-expressing cells	Osteosarcoma	[150]
	TRB + L19-mTNF	Pro-inflammatory cytokine	Fibrosarcoma	[133]
**Physical agents**	TRB + radiotherapy		Lung cancer, colon cancer	[138]
			Advanced soft tissue sarcoma	[139]
			Localized resectable myxoid liposarcoma	[140,141]
			Retroperitoneal leiomyosarcoma	[151]
	TRB + hyperthermia		Osteosarcoma, liposarcoma, synovial sarcoma	[137]

**Table 2 molecules-29-00331-t002:** Combination therapies involving lurbinectedin.

Category	Treatment	Co-Treatment Function	Type of Cancer	Ref.
**Monoclonal antibodies** (**mAb**)	LUR + VE-821 + KU-60019	Anti–ATR (VE-821) Anti–ATM (KU-60019)	Cervical carcinoma, ovarian carcinoma	[122]
**Immune checkpoint inhibitors** (**ICIs**)	LUR + αPD-1 + αCTLA-4		Osteosarcoma, fibrosarcoma, lung cancer, breast cancer	[152]
**Inhibitors**	LUR + irinotecan	Topoisomerase I inhibitor	Ovarian clear cell carcinoma	[154]
			BRCA-mutated platinum-resistant ovarian cancer patient	[153]
	LUR + olaparib	PARP inhibitor	Advanced solid tumours	[157]
	LUR + berzosertib	ATR inhibitor	Small-cell lung cancer	[155]
**Biological agents**	LUR + doxorubicin	Treatment of several sarcomas	Relapsed small-cell lung cancer	[164]
			Leiomyosarcoma, dedifferentiated liposarcoma, myxoid liposarcoma, synovial sarcoma, and desmoplastic small round cell tumour	[41]
			Recurrent advanced endometrial cancer	[163]
	LUR + capecitaine	Treatment of metastatic colorectal cancer (mCRC) and metastatic breast cancer (MBC)	Metastatic breast cancer	[162]
	LUR + paclitaxel	Treatment of several sarcoma	Small cell lung cancer, breast cancer, endometrial cancer	[158]
	LUR + paclitaxel + bevacizumab	Anti–VEGF (bevacizumab)	Epithelial ovarian cancer	[158]
	LUR + cisplatin		Mesothelioma	[160]
	LUR + gemcitabine	Treatment of advanced pancreatic cancer	Advanced solid tumours	[161]
	LUR + 4C9-DM1	Antibody-drug conjugate (ADC) that targets c-Kit	Small cell lung cancer	[165]

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
