# Peer review of "Trabectedin and Lurbinectedin Modulate the Interplay between Cells in the Tumour Microenvironment—Progresses in Their Use in Combined Cancer Therapy"

_molecules, 2024, doi:10.3390/molecules29020331_

Round 1

Reviewer 1 Report

Comments and Suggestions for Authors

Review for the manuscript

“Trabectedin and Lurbinectedin Modulate the Interplay Between Immune Cells: Progresses in Their Use in Combined Therapy”

 The authors present a review manuscript on the two alkaloid structures trabectedin (TRB) and lurbinectedin (LUR) with regard to their therapeutic applicability due to their tumor inhibitory properties. The overarching theme concerns the richness of marine life forms and their typical molecular diversity, which offers great potential for therapeutic purposes in humans. Both molecules and their properties are described in detail, and the findings to date regarding therapeutic approaches and their results are also presented in detail. The tabular presentation of the approaches carried out to date with regard to combination therapy is also well presented, and all details are supported by an extensive bibliography.

Since the underlying topic of molecular structures of marine origin is enormously important due to its potential for human medicine and since the two selected molecules were described very well and in detail, publication of the manuscript in the journal Molecules is recommended.

Before that, however, three minor changes should be made to optimize the manuscript:

 1.       The manuscript is essentially limited to the two molecules TRB and LUR. In order to emphasize the importance of marine life forms, it is recommended that a further table be added in which other important molecules of marine origin are listed with corresponding references, for which a great potential for application in human medicine and especially in oncology has also already been demonstrated. Such a table would further emphasize the importance of the molecules TRB and LUR.
2.       Figure 3 is difficult to read on the left-hand side due to the very small font. This figure should definitely be modified using larger characters, as it otherwise contains very important information that the reader must be able to recognize clearly.
3.       in Figure 4 there is a small typo in the lower half, it should read

"NK-mediated cytotoxicity" instead of "citotoxicity"
After making these minor changes, publication of the manuscript in the journal Molecules is recommended.

 - End of Review –

Author Response

Reviewer 1

“Trabectedin and Lurbinectedin Modulate the Interplay Between Immune Cells: Progresses in Their Use in Combined Therapy”

The authors present a review manuscript on the two alkaloid structures trabectedin (TRB) and lurbinectedin (LUR) with regard to their therapeutic applicability due to their tumor inhibitory properties. The overarching theme concerns the richness of marine life forms and their typical molecular diversity, which offers great potential for therapeutic purposes in humans. Both molecules and their properties are described in detail, and the findings to date regarding therapeutic approaches and their results are also presented in detail. The tabular presentation of the approaches carried out to date with regard to combination therapy is also well presented, and all details are supported by an extensive bibliography.

Since the underlying topic of molecular structures of marine origin is enormously important due to its potential for human medicine and since the two selected molecules were described very well and in detail, publication of the manuscript in the journal Molecules is recommended.

Before that, however, three minor changes should be made to optimize the manuscript

 We sincerely thank the comments that have been addressed by the reviewer and for the effort to improve the interest and quality of the content for the readers.

1.The manuscript is essentially limited to the two molecules TRB and LUR. In order to emphasize the importance of marine life forms, it is recommended that a further table be added in which other important molecules of marine origin are listed with corresponding references, for which a great potential for application in human medicine and especially in oncology has also already been demonstrated. Such a table would further emphasize the importance of the molecules TRB and LUR.
Thank you for the suggestion, it is a great comment, and it makes sense but if we included this table we would have to explain every category and that would be a subject for another review since there is a huge amount of marine-derived by-products with interest in the field of molecular oncology.

  1. Figure 3 is difficult to read on the left-hand side due to the very small font. This figure should definitely be modified using larger characters, as it otherwise contains very important information that the reader must be able to recognize clearly.

Thank you for the valuable comment. We tried our best using larger characters, but they did not fit into the image properly. Instead, we enlarged the image to the maximum width that the format allowed.

3.in Figure 4 there is a small typo in the lower half, it should read "NK-mediated cytotoxicity" instead of "citotoxicity"
Thank you, the mistake has been corrected.

After making these minor changes, publication of the manuscript in the journal Molecules is recommended.

Reviewer 2 Report

Comments and Suggestions for Authors

This manuscript described the progress of the use of trabectedin and lurbinectedin in oncology therapy, especially emphasizing how they modulate immune cells. It is helpful for us to learn the frontier progress. There are some weaknesses as below:

1. According to your title, please subtract other descriptions not related to immune cells. The presented whole text did not focus on the title now. Or you can change your title. Immune cell is not the sole mechanism of action for these drugs to exert their biological activity. Furthermore, please confine your title to cancer therapy.

2. Putting “1.1” in the introduction is not suitable, it should be another independent part.

Author Response

Reviewer 2

This manuscript described the progress about the use of trabectedin and lurbinectedin in oncology therapy,especially emphasized how they modulate immune cells. It is helpful for us to learn the frontier progress. There is some weakness as belows:

1.According to your title, please substract other description not related with immune cells. The presented whole text did not focus on the title now. Or you can change your title. Immune cell is not the sole mechanism of action for these drugs to exert their biological activity. Furthermore, please confine your title to cancer therapy.

 Thank you for the suggestion, after having discussed with all the authors, we decided that the best option involved changing the title to address the matter in a broader sense: “Trabectedin and lurbinectedin modulate the interplay between cells in the tumour microenvironment. Progresses in their use in combined cancer therapy”

 2.Put “1.1”in the introduction is not suitable, it should be another independent part.  

This has been updated, thank you!!

Reviewer 3 Report

Comments and Suggestions for Authors

Adrián Povo-Retana et al  present a review of the literature around the use of Trabectedin (TRB) and Lurbinectedin (LUR) as an antitumoral therapy, the proven mechanisms of action, pathologies where they are used and combined therapies with different immunotherapeutic strategies or specific molecular-targeted inhibitors. In addition, authors provide a hugged amount of evidence on the advantages in terms of antitumoral activity and/or safety profile

of combined therapies that synergise with TRB/LUR. The topic of the manuscript is of interest and also reflects the gaps of knowledge at present. Tables and figures are very clear and give a summary of a variety of studies on the subject. The content of the manuscript is well structured and clearly presents the information in a meaningful way to the reader. 

The literature reviewed is up to date and the manuscript follows a logical order, which makes it easy to read. Overall, the manuscript is very detailed, well written and summarizes the current state of play. 

I just have two minor points, which are detailed below and should be addressed before publication. 

1-Figure 2: It would be helpful to highlight the functional groups mentioned in lines 91-107 as well as the differences between both molecules.

2- Lines 21 and 122: Please, clarify when Lurbinectedin was approved by the FDA-

3- Line 185: Please, define “hTAM”

4- line 266: Please, change “and” for “as”

Comments on the Quality of English Language

Please, check grammar since there are minor issues that need to be solved.

Author Response

Reviewer 3.

Adrián Povo-Retana et al.  present a review of the literature around the use of Trabectedin (TRB) and Lurbinectedin (LUR) as an antitumoral therapy, the proven mechanisms of action, pathologies where they are used and combined therapies with different immunotherapeutic strategies or specific molecular-targeted inhibitors. In addition, authors provide a hugged amount of evidence on the advantages in terms of antitumoral activity and/or safety profile

of combined therapies that synergise with TRB/LUR. The topic of the manuscript is of interest and also reflects the gaps of knowledge at present. Tables and figures are very clear and give a summary of a variety of studies on the subject. The content of the manuscript is well structured and clearly presents the information in a meaningful way to the reader. 

The literature reviewed is up to date and the manuscript follows a logical order, which makes it easy to read. Overall, the manuscript is very detailed, well written and summarizes the current state of play. 

I just have two minor points, which are detailed below and should be addressed before publication. 

We would like to thank the reviewer for the positive comments and the careful revision of the manuscript.

1.Figure 2: It would be helpful to highlight the functional groups mentioned in lines 91-107 as well as the differences between both molecules.

We would like to thank the reviewer for the observation: The differences between both molecules are located in the upper part of the image. To avoid potential confusion in highlighting atoms we described the differences in the functional groups mentioned in lines 91-107 and circled these differences in blue for TRB and red in LUR in figure two. We indicated this addition in the main body of the text. We think that with this simple explanation the readers are better oriented.

2.Lines 21 and 122: Please, clarify when Lurbinectedin was approved by the FDA-

Thank you for the comment!! It was approved in 2020.

3.Line 185: Please, define “hTAM”

hTAM stands for human Tumour-Associated Macrophage it has been included in that line, thank you!!

4.line 266: Please, change “and” for “as”

Thank you for detecting the typing mistake. It has been corrected.
